# Tensorial stress-plastic strain fields in α - ω Zr mixture, transformation kinetics, and friction in diamond-anvil cell

Valery I. Levitas [1,2,3,5] ✉, Achyut Dhar [1,5] ✉ & K. K. Pandey[4]

Various phenomena (phase transformations (PTs), chemical reactions, microstructure evolution, strength, and friction) under high pressures in diamond-anvil cell are strongly affected by fields of stress and plastic strain tensors. However, they could not be measured. Here, we suggest coupled experimental-analytical-computational approaches utilizing synchrotron X-ray diffraction, to solve an inverse problem and find fields of all components of stress and plastic strain tensors and friction rules before, during, and after α-ω PT in strongly plastically predeformed Zr. Results are in good correspondence with each other and experiments. Due to advanced characterization, the minimum pressure for the strain-induced α-ω PT is changed from 1.36 to 2.7 GPa. It is independent of the plastic strain before PT and compression-shear path. The theoretically predicted plastic strain-controlled kinetic equation is verified and quantified. Obtained results open opportunities for developing quantitative high-pressure/stress science, including mechanochemistry, synthesis of new nanostructured materials, geophysics, astrogeology, and tribology.

In static high-pressure studies, high pressures are generated by compression, with large very- heterogeneous elastoplastic deformations, of a thin sample down to 6–20 microns in a diamond-anvil cell (DAC)[1–7]; see Fig. 1a. The same happens when the pressure-transmitting medium solidifies (Supplementary Fig. 12). We will focus here on stresses and plastic strains averaged over the poly-crystalline aggregate rather than in individual grains. The most advanced characterization of the pressure conditions in a sample is based on determining the radial distribution of pressure averaged over the sample thickness using the volume of a crystal cell measured by X-ray diffraction (XRD) and equation of state (EOS) determined under hydrostatic conditions[4,5,8–11]. However, EOS for hydrostatic and non-hydrostatic loadings are quite different[12–15]. More importantly, for the XRD beam along the symmetry axis of the DAC (axial XRD), crystallographic planes that are almost parallel to

the beam contribute to the measured XRD patterns only, and axial elastic strain $\bar{E}_{0,zz}$ and consequently stress $\bar{\sigma}_{zz}$ do not contribute to the pressure, leading to large error (bar over the field variables means averaged over the sample thickness). In addition, numerous physical, chemical, geological, and mechanical problems and phenomena are related to knowledge of the fields of all components of the stress, elastic, and plastic strain tensors in materials compressed in DAC[1–7,11–13,16–31]. For example, contact friction shear stress between diamond and sample/gasket is responsible for generating high pressure and is the key boundary condition for simulation of the processes in DAC;[1,4–6,21,32–39] however, the friction rules are unknown. It is known that phase transformations (PTs) and chemical reactions strongly depend on the non-hydrostatic stresses and plastic strains[11,16,20,22,23,26–31,40,41], even within different pressure-transmitting media at relatively low pressure[24,42]. New types, namely plastic

[1]Department of Aerospace Engineering, Iowa State University, Ames, IA 50011, USA. [2]Department of Mechanical Engineering, Iowa State University, Ames, IA 50011, USA. [3]Ames National Laboratory, Division of Materials Science and Engineering, Ames, IA 50011, USA. [4]High Pressure & Synchrotron Radiation Physics Division, Bhabha Atomic Research Centre, Bombay, Mumbai 400085, India. [5]These authors contributed equally: Valery I. Levitas, Achyut Dhar.
✉e-mail: vlevitas@iastate.edu; adhar@iastate.edu

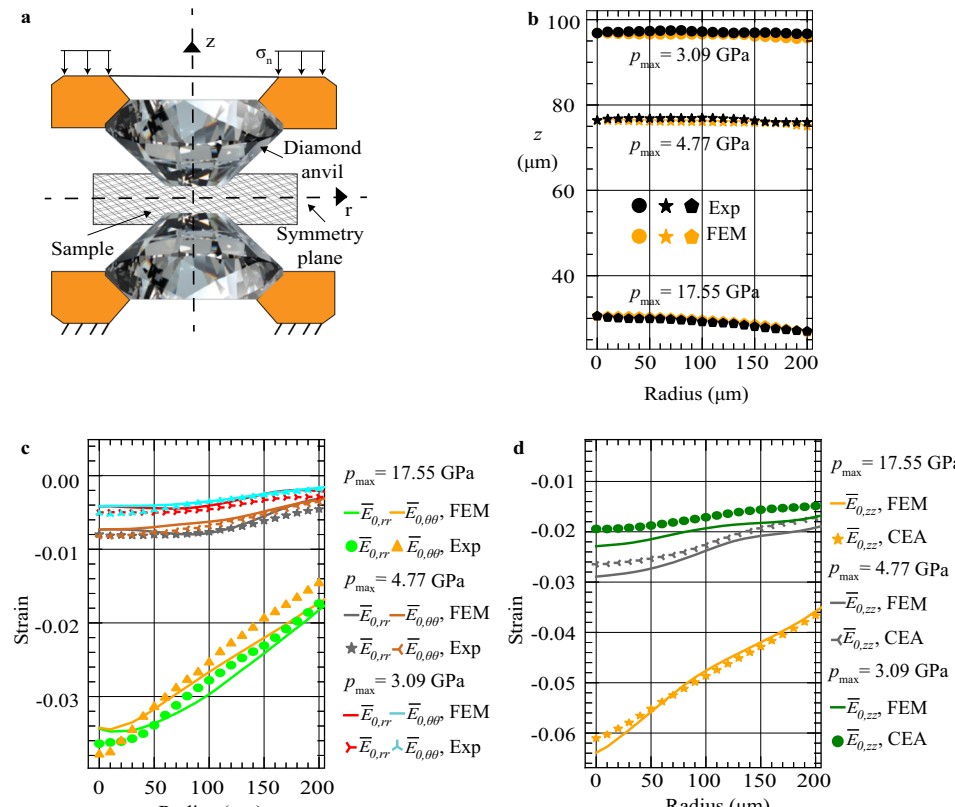

**Fig. 1 | Distributions of different strains in DAC. a** DAC schematic. **b** The sample thickness profiles from the X-ray absorption and FEM. **c** Comparison of experimental and FEM results for distributions of elastic radial $\bar{E}_{0,rr}(r)$ and the hoop $\bar{E}_{0,\theta\theta}(r)$ strains in a mixture averaged over the sample thickness. **d** Comparison of FEM and CEA distributions of elastic axial strains $\bar{E}_{0,zz}$.

strain-induced PTs and reactions were formulated and are under intense studies[11,23,40,43]. Plastic strain can reduce PT pressure in comparison with quasi-hydrostatic experiment from more than 52.5 to 6.7 GPa for rhombohedral to cubic BN[27], from 55 to 5.6 GPa for hexagonal to superhard wurtzitic BN[20], and from 70 to 0.7 GPa for graphite to cubic diamond[22], which may lead to new technologies. Also, the plastic strain may lead to new nanostructured phases that could not be obtained under hydrostatic conditions, substitute time-dependent kinetics with fast plastic strain-dependent kinetics, and substitute reversible PTs with irreversible PTs that allow one to use retrieved phases in engineering applications[11,22,23,27,28,40]. Severe plastic deformation with high-pressure torsion[44] is an example of the realization of such technologies. Strain-induced PTs under high pressure also occur during surface processing (polishing, turning, scratching, etc.) of strong brittle semiconductors and ceramics and are utilized for developing regimes of ductile machining[45]. Again, since stress and plastic strain tensors are not measurable, quantitative studies of these phenomena are impossible.

The paper that claims measurement of radial distribution of all components of the stress tensor is ref. 46. However, this measurement is performed in a diamond culet rather than in a sample, which gives boundary normal and shear stresses at the sample boundary only instead of full stress and plastic strain tensor fields in the entire sample. Since there was a problem in the precise measurement of the shear stress, finite element method (FEM) simulation was utilized to supplement the experiment. In ref. 35, all tensorial fields are determined in the polycrystalline W sample; however, input data are the radial pressure distributions determined using EOS. Measured displacements of material particles at the contact surface with diamond[37] represent important boundary conditions, which still were not connected to FEM simulations.

Here, we develop coupled experimental-analytical (CEA) and CEA-FEM approaches for solving an inverse problem of determining fields of all stress, elastic and plastic strain components (in each phase and in the mixture), and friction shear stress before, during, and after α-ω PT in commercially pure polycrystalline Zr, see flowchart in Supplementary Fig. 1. Importantly, to exclude the effect of strain hardening, change in grain size and dislocation density, and their effect on the thermodynamics and kinetics of PT, we have strongly preliminary deformed Zr until its hardness does not change[11,34]; grain size and dislocation density in pure α- and ω-Zr do not change with further straining as well[47]. This is a crucial step to make the problem solvable. Next, based on limited access of the beam for axial geometry, determined fields of XRD patterns, and texture of both phases, we concluded that the most informative and precise approach is to determine, through postprocessing, distributions of the elastic radial $\bar{E}_{0,rr}(r)$ and hoop $\bar{E}_{0,\theta\theta}(r)$ strains in α and ω phases. Sample thickness profile and pressure-dependence of the yield strength of phases are determined using X-ray absorption[4,5,11] and broadening of X-ray peaks[11,48]. The CEA approach is developed to determine friction shear stress and all components of stress and elastic strain tensors in each phase and mixture of phases based on XRD measurements. Next, using found friction stress, detailed FEM modeling, and simulation are performed that determine all components of stress and plastic strain tensors, which completes the problem in the CEA-FEM approach. Remarkably, the results of analytical and FEM solutions for all stresses are in good correspondence. Distributions of $\bar{E}_{0,rr}(r), \bar{E}_{0,\theta\theta}(r)$ and sample thickness profiles calculated with FEM perfectly correspond to experiments. Obtained pressure distribution differs significantly from that using the EOS-based method. The corrected minimum pressure for the strain-induced α - ω PT is $p_\varepsilon^d$ =2.70 GPa vs. 1.36 GPa based on the EOS method. Still, it is smaller than under hydrostatic loading by a

factor of 2 (and smaller than the phase equilibrium pressure by a factor of 1.3); it is found to be independent of the plastic strain tensor and its path, in particular, of the compression-shear strain path. The theoretically predicted plastic strain-controlled kinetic equation was verified and quantified; it is independent of the plastic strain at pressures below $p_\varepsilon^d$ and the pressure-plastic strain loading path. Based on our texture analysis (see Supplementary Notes), $c$ axis of $\alpha$-Zr is predominantly aligned along the loading direction; however, $c$ axis of $\omega$-Zr is predominantly aligned along the radial direction. In addition, based on the analysis of the XRD peak broadening[48], we obtained the yield strength of $\alpha$-Zr $\sigma_y^\alpha = 0.82 + 0.190p$ (GPa) and $\omega$-Zr $\sigma_y^\omega = 1.66 + 0.083p$ (GPa) (Supplementary Fig. 9).

## Results

Results for three different loadings identified by averaged over the thickness pressure in the mixture $\bar{p}$ at the symmetry axis (i.e., maximum pressure $p_{\max}$) are presented in Fig. 1. The loadings with $p_{\max} = 3.09$ GPa is for the almost pure $\alpha-Zr$ phase, for $p_{\max} = 4.77$ GPa is for the mixture of $\alpha$- and $\omega$-Zr, and for $p_{\max} = 17.55$ GPa is for the pure $\omega-Zr$ phase. The corresponding volume fraction profiles $c(r)$ are shown at the bottom of Fig. 2. The sample thickness profiles from the X-ray absorption and FEM are in good correspondence (Fig. 1b). There is also good correspondence between experiments and FEM distributions of $\bar{E}_{0,rr}(r)$ and $\bar{E}_{0,\theta\theta}(r)$, as well as the closeness of $\bar{E}_{0,rr}(r)$ to $\bar{E}_{0,\theta\theta}(r)$ (Fig. 1c). Both experimental verifications of FEM results represent nontrivial validation of the model and simulations; thus, all FEM fields presented below represent reality, even if they cannot be directly measured. Figure 1d shows good correspondence between FEM and CEA distributions of the elastic axial strains $\bar{E}_{0,zz}(r)$, which is a part of the validation of the analytical model.

The comparison of different radial stress distributions obtained with FEM and CEA is given in Fig. 2. The practical coincidence of the contact friction stress $\tau_c$ from CEA and FEM is not surprising because the field $m(r)$ from the analytical solution is used in FEM as the boundary condition. All stresses smoothly increase from the edge of a culet to the sample center. For all three loadings, there is a very good, and for some stresses, excellent correspondence between the analytical and the FEM results. In addition, FEM distributions of $\sigma_{zz}^c$ at the contact surface and $\sigma_{zz}^{sp}$ at the symmetry plane do not differ essentially, which supports the assumption in the analytical model that $\sigma_{zz}$ is independent of $z$. It can also be seen that $\sigma_{rr} \approx \sigma_{\theta\theta}$ from FEM at the symmetry plane, contact surface, and averaged over the thickness, which justifies the assumption $\sigma_{rr} = \sigma_{\theta\theta}$ made in the analytical model. Comparison of 2D stress contours obtained with CEA and FEM approaches is presented in Supplementary Fig. 11. Maximum values of the same stresses are practically the same, character of changes of all stresses is also the same, i.e., correspondence is good. Thus, despite the simplicity and numerous assumptions, the analytical model describes well stress fields from FEM, and can be used for analysis and interpretation of experiments. Friction shear stress is essentially lower than the yield strength for all pressures. That means that the known method[6,21,32] to determine the yield strength in shear based on the equilibrium Eq. (59) (see Supplementary Notes) and assumption $m = 1$ does not work. From the edge toward the center, friction stress grows, reaches the maximum and then reduces to zero at the center of a sample due to symmetry conditions.

Friction laws can be formalized with the following equations (Supplementary Fig. 8):

$$\left(\frac{\tau_c}{\tau_y^c}\right)_\omega = 0.186 + 0.018p^c \qquad \text{for } 11.1 \le p^c \text{(GPa)} \le 15.0; \qquad 60 \le r\,(\mu m) \le 140$$

$$\left(\frac{\tau_c}{\tau_y^c}\right)_{\alpha+\omega} = -0.179 + 0.241p^c \qquad \text{for } 2.7 \le p^c \text{(GPa)} \le 3.7; \qquad 130 \le r\,(\mu m) \le 200$$

$$\left(\frac{\tau_c}{\tau_y^c}\right)_\alpha = -1.282 + 0.722p^c \qquad \text{for } 2.0 \le p^c \text{(GPa)} \le 2.45; \qquad 130 \le r\,(\mu m) \le 190 .$$

$$(1)$$

The radial distributions of averaged through thickness pressure in $\alpha$-Zr $\bar{p}^\alpha$ are shown in Fig. 3a. They are obtained using a developed

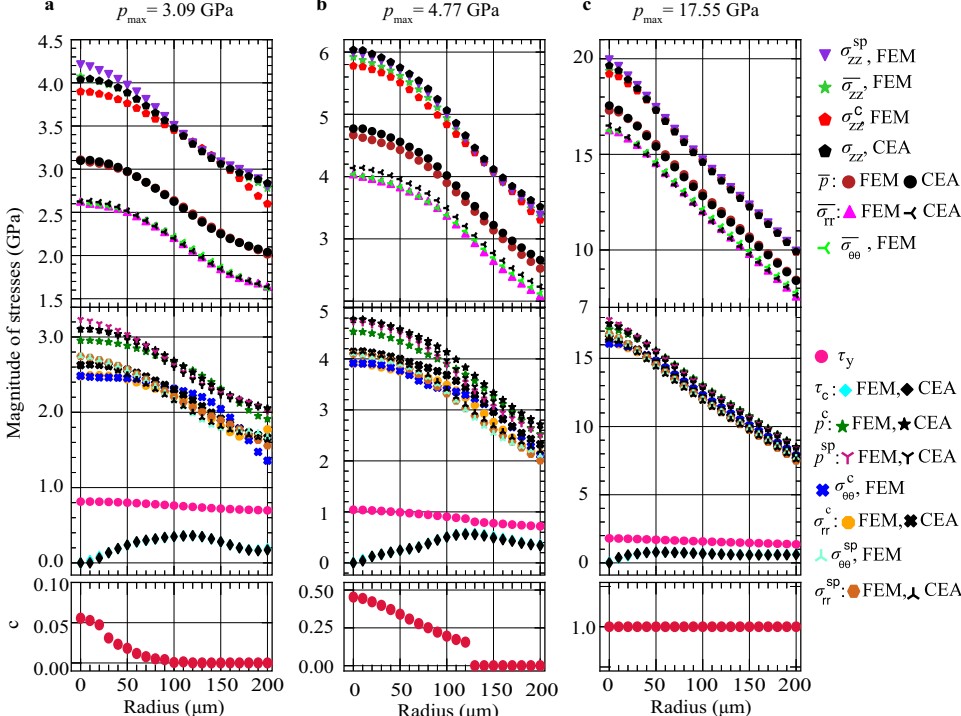

**Fig. 2 | Comparison of different radial stress distributions obtained with FEM and analytically. a** Results for almost pure $\alpha-Zr$ at $p_{\max} = 3.09$ GPa. **b** Results for mixture of $\alpha$- and $\omega$-Zr at $p_{\max} = 4.77$ GPa. **c** Results for pure $\omega-Zr$ at $p_{\max} = 17.55$ GPa. Distributions of the volume fraction $c(r)$ for the corresponding loadings are shown at the bottom of the figure. Subscripts 'c' and 'sp' mean contact surface and symmetry plane, respectively.

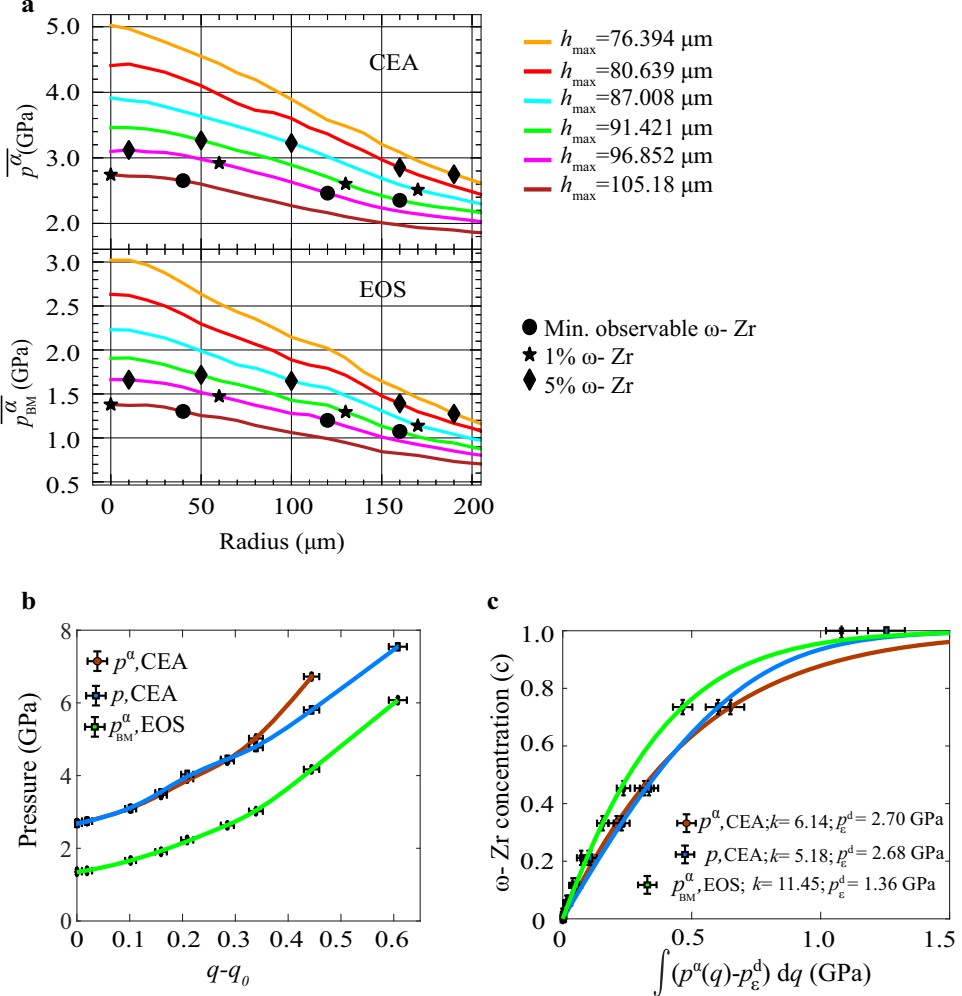

**Fig. 3 | Kinetics of α - ω PT. a** The radial distribution of $\bar{p}^\alpha$ in α-Zr obtained using developed CEA based on experimentally measured radial $\bar{E}_{0,rr}(r)$ and the hoop $\bar{E}_{0,\theta\theta}(r)$ strains (top) and hydrostatic EOS based on experimentally measured elastic volumetric strain, for the same sample thicknesses. Three types of markers are superposed on these curves corresponding to $c$ = 0.05, 0.01, and minimum observable traces of ω − Zr. **b** Loading pressure-accumulated plastic strain paths. Error in the estimation of $q$ is based on the error in estimation of thickness which is 1 micron. Error in the estimation of pressure of 0.2 GPa is based on the error in estimation of lattice parameters. **c** Corresponding kinetics of evolution of ω − Zr concentration based on pressures in the parent α-Zr obtained using CEA and EOS, as well as averaged over the mixture pressure obtained with the CEA; lines correspond to Eq. (2). Here $q_o$ is the accumulated plastic strain at the beginning of PT. Error in $\int(p(q) - p_\varepsilon^d)dq$ is evaluated based on the convolution of errors in estimating the pressure and the $q$. Error in the estimation of concentration of 0.05 is based on fitting XRD with 2 phases.

approach (top) and using hydrostatic (bottom), for the same sample thicknesses. It is evident that the suggested CEA approach to post-process X-ray measurements leads to pressures higher by a factor of 1.7–2.0 or by 1.4–2.0 GPa (i.e., at the level of the yield strength $\sigma_y$ at the corresponding pressure) than those obtained by traditional utilization of EOS. This is a quite significant correction that should be applied to all previous publications in pressure measurements based on EOS[4,5].

The obtained corrections also lead to a reinterpretation of the kinetics of α - ω PT in comparison with that in[11]. The strain-controlled kinetic equation derived in ref. [23] and simplified for Zr in ref. [11] is

$$\frac{dc}{dq} = k(1-c)\frac{p(q)-p_\varepsilon^d}{p_h^d - p_\varepsilon^d} \rightarrow c = 1 - \exp\left(\frac{-k}{(p_h^d - p_\varepsilon^d)}\int(p(q)-p_\varepsilon^d)dq\right) \tag{2}$$

where $p$ is pressure either in mixture or in α-Zr ($\bar{p}^\alpha$), $q$ is the accumulated plastic strain, $p_h^d$ is the pressure for initiation of pressure-induced PT under hydrostatic loading, $p_\varepsilon^d$ is the minimum pressure for initiation of the plastic strain-induced PT, $p(q)$ is the loading path. To quantify Eq. (2), experimental points at the center of the sample are

used, where unidirectional compression is realized, $q = \ln(h_o/h)$, where $h_o$ and $h$ are the initial and current sample thicknesses at the center. For strongly plastically pre-deformed Zr we found that $p_h^d = 5.4$ GPa. Figure 3b shows the experimental loading path $p(q)$ based on different pressures: pressures in the parent α-Zr obtained using an analytical model and EOS, as well as averaged over the mixture pressure obtained with an analytical model. The loading path with the analytical model is shifted up with respect to the EOS-based model by 1.3–2.0 GPa. To detect the initiation of PT, three types of markers are superposed on the pressure distribution curves corresponding to $c = 0.05$, 0.01, and minimum observable traces of ω − Zr. The minimum pressure for initiation of the plastic strain-induced PT is determined by extrapolating $p$-$c$ results to $c = 0$ at the sample center, which gives $p_\varepsilon^d = 2.70$ GPa with CEA approach instead of $p_\varepsilon^d = 1.36$ GPa based on EOS. Thus, the developed method led to an essential increase in the minimum PT pressure. Still, it is two times lower than under hydrostatic conditions and lower than the phase equilibrium pressure of 3.4 GPa. Note that with the EOS method, $p_\varepsilon^d$ here for commercially pure Zr is slightly higher than 1.2 GPa for ultra-pure Zr in ref. [11]. It is important that for both methods of pressure determination, all three

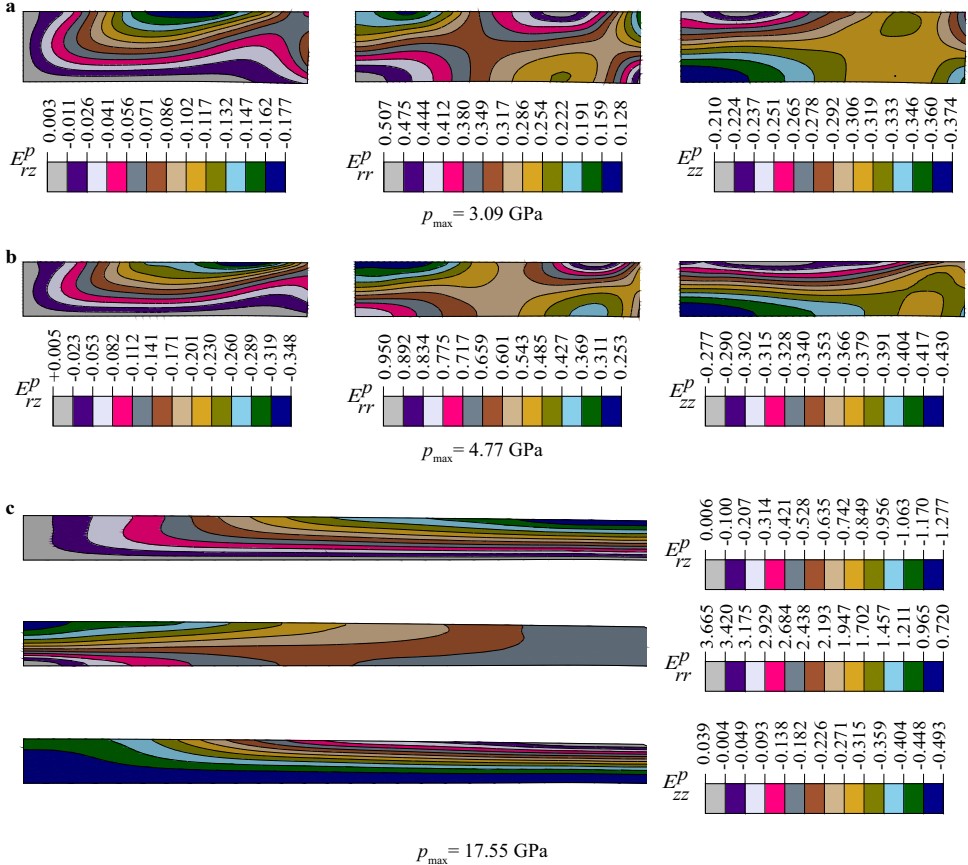

**Fig. 4 | Distributions of components of Lagrangian plastic strains in a sample for three loadings. a** Results for almost pure $\alpha - Zr$ at $p_{max} = 3.09$ GPa. **b** Results for mixture of $\alpha$- and $\omega$-Zr at $p_{max} = 4.77$ GPa. **c** Results for pure $\omega - Zr$ at $p_{max} = 17.55$ GPa.

types of markers in Fig. 3a show very close values for different radii, i.e., the minimum PT pressure is practically independent of $r$. However, the plastic strain tensor and its path are very different for different radii. At the center, unidirectional compression without shear takes place while with increasing radius shear strain grows. Consequently, $p_\varepsilon^d$ is independent of the plastic strain tensor and its path, in particular, of compression/shear plastic strain state and its path. This means that there is no advantage of shear deformation mode in promoting PTs, physical mechanisms are the same for PT under plastic compression and under shear, and PT processes under compression in DAC and torsion in rotational DAC require the same experimental characterization and theoretical treatment. However, rotational DAC allows to independently control pressure and plastic strain and produces PT up to completion close to $p_\varepsilon^d$, which is also important for technologies of plastic strain and defect-induced material synthesis at relatively low pressure.

Experimental points for PT kinetics based on three different pressures are well described by Eq. (2) with different $p_\varepsilon^d$ and kinetic coefficients $k$, see Fig. 3c. This validates Eq. (2) for a quite nontrivial loading path. While the difference between kinetic curves with pressures based on the CEA approach and EOS does not look drastic, this is due to the choice of independent variables along the horizontal axis, different for different cases. Quantitatively, the CEA method led not only to an increase in $p_\varepsilon^d$ by a factor of 2 but also to a decrease in kinetic coefficient $k$ from 11.45 to 6.14, i.e., correction is very significant. Since this correction did not change the conclusion that $p_\varepsilon^d$ is independent of compression/shear plastic strain state and its path, we assume that other conclusions from[11], that Eq. (2) is independent of the magnitude of plastic strain $q_o$ below $p_\varepsilon^d$ and of $p - q$ loading path, are valid as well.

Fields of all components of Lagrangian plastic strain tensor found with the CEA-FEM approach are presented in Fig. 4. Very heterogeneous and nontrivial distributions are observed, caused by heterogeneous contact friction. These fields will be used in future work, which will include simulation of the strain-induced PT as well, to derive, calibrate, and validate a more precise kinetic equation for strain-induced PT.

To summarize, we developed coupled CEA and CEA-FEM approaches that allow us to solve the inverse problem reinterpret X-ray diffraction measurements, and determine pressure and tensorial stress-plastic strain fields in each phase and mixture, as well as contact friction rules in a DAC before, during, and after $\alpha$-$\omega$ transformation in Zr. Good correspondence of the elastic radial $\bar{E}_{0,rr}(r)$ and hoop $\bar{E}_{0,\theta\theta}(r)$ strains and the sample thickness profile between FEM and experiments and the stress tensor fields between the CEA approach and FEM validates the developed approach. Due to advanced characterization, the minimum pressure $p_\varepsilon^d$ for the strain-induced $\alpha$-$\omega$ PT in Zr is changed from 1.36 to 2.7 GPa and the kinetic coefficient $k$ is reduced from 11.45 to 6.14, i.e., correction is very significant. Still, $p_\varepsilon^d$ is 2 times lower than under hydrostatic conditions and lower than the phase equilibrium pressure of 3.4 GPa. The found independence of $p_\varepsilon^d$ of the plastic strain tensor and its path (in particular, compression-shear path) means basic equivalence of PT processes under compression in DAC and torsion in rotational DAC. However, rotational DAC allows to independently control pressure and plastic strain and produces PT up to completion at pressures close to $p_\varepsilon^d$, which is important for technologies of plastic strain and defect-induced material synthesis at relatively low pressure. Since our pressure correction did not change the conclusion that $p_\varepsilon^d$ is independent of compression/shear plastic strain state and its path, it is highly probable that other conclusions from[11], that kinetic Eq. (2) is

independent of the magnitude of plastic strain $q_o$ below $p_\varepsilon^d$ and of $p$-$q$ loading path, are valid as well.

The obtained results in this work open opportunities for developing quantitative high-pressure/stress science. The same methods are applicable for other material systems (including gasket materials), for sample-gasket systems, without and with hydrostatic medium, after its solidification, and can be extended for processes in rotational DAC (Supplementary Fig. 12). They can significantly improve the accuracy of pressure field determination and characterization of all processes studied under pressure: physical, chemical, biological, geophysical, and others. Finding fields of stress tensor components (which can be done using analytical model) will allow quantifying their effect on the processes under study, instead of referring to the qualitative effect of pressure-transmitting media and "non-hydrostatic" stresses. Finding fields of plastic strain tensor components, that cannot be measured, will allow us to quantitatively study plastic strain-induced PTs and chemical reactions and initiate quantitative high-pressure mechanochemistry. This may lead to new technologies of plastic strain and defect-induced material synthesis at relatively low pressure, in particular, for diamond[22] and cubic BN[20,27], initiation of high-pressure tribology, explanation of deep-focus earthquakes[29], the appearance of microdiamond in the low pressure and temperature Earth's crust[22], and mechanochemical origin of life in the icy crust of solar system's moons and planets[30,31]. Our method generates big data from single experiments/simulation, which can be utilized for machine learning-based development and calibration of the corresponding constitutive equations. Thus, the main challenge, namely, strong heterogeneity of all fields, can be transformed into an opportunity.

## Methods

### CEA approach

Two mechanical equilibrium equations for the axisymmetric model in radial $r$ and axial $z$ directions, the pressure-dependent *von-Mises* yield equation for isotropic perfectly plastic polycrystal $\left(\frac{3}{2} S_{ij} S_{ij}\right)^{0.5} = \sigma_y = (1 - c)\sigma_y^\alpha(p) + c\sigma_y^\omega(p)$, and the assumption that radial stress $\sigma_{rr}$ is equal to azimuthal stress $\sigma_{\theta\theta}$, i.e. $\sigma_{rr} = \sigma_{\theta\theta}$, form 4 equations with four unknown stresses, $\sigma_{rr}$, $\sigma_{\theta\theta}$, axial $\sigma_{zz}$, and shear stress $\tau = \tau_{rz}$. Here, $\sigma_y$ is the yield strength in compression of the mixture, $c$ is the volume fraction of ω-Zr, and $S_{ij}$ are components of the deviatoric stress tensor in the mixture. An approximate analytical solution to this statically determined system of equations is found by modifying the Prandtl solution for the plane strain (see Supplementary Notes, Section 2). However, it depends on unknown contact friction shear stress $\tau_c = m\tau_y(p, c)$, where $\tau_c = \frac{\sigma_y(p,c)}{\sqrt{3}}$ is the yield strength in shear of the mixture and $m$ is the factor to be determined. To find the distribution of the friction stress in terms of measured contribution of elastic strain averaged over the sample thickness $\bar{E}_{0,rr}(r) \approx \bar{E}_{0,\theta\theta}(r)$, i.e., in terms of $0.5(\bar{E}_{0,rr}(r) + \bar{E}_{0,\theta\theta}(r))$, the equations derived in (see Supplementary Notes, Section 2) are used.

The modified Hooke's law for hydrostatically pre-stressed properly oriented α- and ω-Zr single crystals with determined pressure-dependent elastic moduli is used to determine stresses in each phase and mixture (see Supplementary Notes). Then both Hooke's law and stress fields from the modified Prandtl solutions are averaged over the sample thickness for each $r$. The *Reuss* hypothesis is used that stresses in a mixture of all α- and ω-Zr single crystals in the representative volume and in polycrystalline aggregate (that participate in the modified Prandtl solution) are the same. This hypothesis appears to work well due to the highly textured polycrystalline aggregate. Finally, the simplified mechanical equilibrium equation averaged over the sample thickness is utilized, to determine the contact friction shear stress. After the solution of the obtained nonlinear system of algebraic/trigonometric equations for $m(r)$ and $\bar{p}$, all components of the fields of stress and elastic strain tensors in each phase and mixture of phases are obtained analytically, but plastic strains are unknown.

### FEM modeling and simulations

A large elastoplastic strain model for mixture of α- and ω-Zr using the mixture rule for all properties is advanced (see Supplementary Notes). The evolution of the field of the measured volume fraction of ω-Zr $c(r)$ and corresponding isotropic transformation strain are introduced homogeneously along the $z$ coordinate. Obtained analytically evolution of the field $m(r)$ is used as the boundary condition for the contact problem in the culet portion. At the inclined portion of the sample-anvil contact surface, the contact shear stress is determined by the minimum between $\tau_c = m\tau_y(p)$, with the value of $m$ at the culet-inclined surface boundary, and Coulomb friction. The elastic constitutive response of polycrystalline Zr is modeled using 3rd-order Murnaghan potential. Associated flow rule in deviatoric stress space is used along with plastic incompressibility. The elastic response of the diamond is modeled using 4th order elastic potential for cubic crystal averaged over azimuthal direction to keep the axial symmetry.

### Materials

The material studied in the paper is the same as was used by Zhilyaev et al.[49], purchased from Haines and Maassen (Bonn, Germany), i.e., commercially pure (99.8%) α-Zr (Fe: 330 ppm; Mn: 27 ppm; Hf: 452 ppm; S: <550 ppm; Nd: <500 ppm). The sample slab with initial thickness of 5.25 mm was cold rolled down to ~165 μm to obtain plastically pre-deformed sample with saturated hardness. Vickers microhardness test method was used to characterize the hardness of the sample at several steps during cold rolling. A 3 mm diameter disk was punch cut from thus obtained thin rolled sheet for unconstrained compression experiments in DAC. For hydrostatic compression experiments, small specks of ~20 μm size were chipped off from the plastically pre-deformed sample using the diamond file.

The hydrostatic high-pressure X-ray diffraction measurements were carried out using the same DAC to estimate equation of state, bulk modulus, and its pressure derivative at ambient pressure for this sample. For these experiments, small Zr specks of ~20 μm size, as already mentioned, were loaded in sample chamber along with silicone oil and copper chips as pressure-transmitting medium and pressure marker respectively. The sample chamber was prepared by drilling a hole of ~250 μm diameter in steel gaskets pre-indented using diamond anvils from initial thickness of ~250 μm to ~50 μm. Hydrostatic high-pressure experiments were carried out in small pressure steps of ~0.2 GPa up to a maximum pressure of 16 GPa.

### Experimental techniques and methodology

Unconstrained plastic compression experiments were carried out prescribing different compression loads to plastically pre-deformed Zr sample loaded in DAC without any constraining gasket. The sample was subjected to axial loads of 50 N, 100 N, 150 N, 170 N, 190 N, 210 N, 230 N, 250 N, 270 N, 290 N, 310 N, 330 N, 350 N, 400 N, 450 N, 500 N, 550 N, 600 N, 650 N, 700 N, 750 N, 800 N, 850 N, 900 N, 950 N, and 1000 N.

In situ XRD experiments were performed at 16-BM-D beamline at HPCAT sector at Advanced Photon Source employing focused monochromatic X-rays of wavelength 0.3096(3) Å and size ~6 μm × 5 μm (full-width at half maximum). At each load-condition, the sample was radially scanned over the entire culet diameter (500 μm) in steps of 10 μm, and 2D diffraction images were recorded at Perkin Elmer flat panel detector. At each load step, X-ray absorption scan was also recorded in same 10 μm steps to obtain thickness profile of sample under given load condition.

2D diffraction images were converted to a 1D diffraction pattern using FIT2D software[50,51] and subsequently analyzed through Rietveld refinement[52,53] using GSAS II[54] and MAUD[55] software for obtaining

lattice parameters, phase fractions, and texture parameters of both α and ω phases of Zr. Based on the different angular dependence of the gain size and microstrain contributions to the diffraction peak broadening, they can be separated. The whole powder pattern fitting using the modified Rietveld method (as implemented in MAUD software[56]) was utilized, which takes texturing and stress anisotropy into account.

In axial geometry (i.e., when the incident X-ray beam is directed along $z$ axis) (Supplementary Figs. 1a and 10), the diffraction condition is satisfied mostly for those crystallographic planes that are nearly parallel (plane normal perpendicular) to the load axis. Hence the observed shifts in diffraction peaks can be practically used to estimate strains in radial and azimuthal directions viz. $\bar{E}_{0,11} = \bar{E}_{0,rr}$ and $\bar{E}_{0,22} = \bar{E}_{0,\theta\theta}$ averaged over the sample thickness. Ideally, the angle between the load axis and diffraction vector $\psi$ should be equal to $90°$ to estimate these strain components. However, since this is not possible in axial geometry, we can use the diffraction peak with smallest diffraction angle, $\theta$. In our experiments for α-Zr, (100) diffraction peak appears at $\theta = 3.18°$ for used X-rays ($\lambda = 3.1088$ Å) at ambient pressure. This corresponds to $\psi = 86.82°$ and can be used for estimation of strain components $\bar{E}_{0,rr}$ and $\bar{E}_{0,\theta\theta}$. Note that (100) peak corresponds to '$a$' lattice parameter because $c$-axis of α-Zr is predominantly aligned along the loading direction as per our texture analysis.

For ω-Zr, (001) diffraction peak appearing at $\theta = 2.85°$ ($\psi = 87.15°$) can be used for estimation of strain components $\bar{E}_{0,rr}$ and $\bar{E}_{0,\theta\theta}$. The (001) peak of ω-Zr corresponds to '$c$' lattice parameter and as per texture analysis, $c$ axis of ω-Zr is predominantly perpendicular to the loading direction of DAC.

Thus, strain components $\bar{E}_{0,rr}$ and $\bar{E}_{0,\theta\theta}$ for α and ω phases of Zr have been obtained for each loading condition at each scanning position using the following equations:

For α-Zr:

$$\bar{E}_{0,rr} = 0.5\left(\left(a/a_0\right)^2 - 1\right) \text{ using } \phi = 0^o \text{ sector of (100) diffraction ring;}$$

$$\bar{E}_{0,\theta\theta} = 0.5\left(\left(a/a_0\right)^2 - 1\right) \text{ using } \phi = 90^o \text{ sector of (100) diffraction ring;}$$

For ω-Zr:

$$\bar{E}_{0,rr} = 0.5\left(\left(c/c_0\right)^2 - 1\right) \text{ using } \phi = 0^o \text{ sector of (001) diffraction ring;}$$

$$\bar{E}_{0,\theta\theta} = 0.5\left(\left(a/a_0\right)^2 - 1\right) \text{ using } \phi = 90^o \text{ sector of (001) diffraction ring.}$$

Finally, diffraction data at the symmetry axis for all load conditions were used for quantitative analysis of the kinetics of plastic strain-induced α−ω phase transition in Zr, like in (ref. 11). For this purpose, the pressure in α-Zr and volume fraction of ω-Zr were estimated as a function of accumulated plastic strain $q$. At the symmetry axis, material experiences a unidirectional compression, for which $q = \ln\left(h_0/h\right)$, where $h_o$ is the initial thickness of the sample in DAC and $h$ is the current thickness.

## Data availability

Source data are provided with this paper.

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

## Acknowledgements

Support from NSF (CMMI-1943710, DMR-2246991, and XSEDE MSS170015), and Iowa State University (Vance Coffman Faculty Chair Professorship and Murray Harpole Chair in Engineering) is greatly appreciated.

## Author contributions

V.I.L. conceived the study, supervised the project, developed theoretical models, and secured funding. A.D. performed the simulations. A.D. and V.I.L. prepared the initial manuscript. K.K.P. performed experiments and collected and postprocessed data. All authors contributed to discussions of the data and to the writing of the manuscript.

## Competing interests

The authors declare no competing interests.
