## [Peer Review File · Nature Communications]

Tensorial stress-plastic strain fields in α - ω Zr mixture, transformation kinetics, and friction in diamond anvil cellREVIEWER COMMENTS

Reviewer #1 (Remarks to the Author):

This is a very 'solid' report that attempts to bring several simulations and experimental data together to answer a very long standing question regarding the interplay between stresses, plastic strain and pressure measurement. The authors apply it to the long studied (and probably the problem that raises questions with every study) alpha-omega transition in Zr. The current manuscript also tries to evaluate a plethora of studies that are arising out of shear devices like the rotational diamond anvil cell (RDAC). In fact, this study highlights the pitfalls of studies with RDAC that can end up being highly irreproducible since improper characterization of the stress fields leads to improper estimation of pressure. This is highlighted with simulations and careful experimental data on the radial pressure distributions at various peak pressures that are validated by FEM calculations. I have no hesitation in recommending this for publication as is. This research would be followed by many high pressure researchers from as diverse a field as rock mechanics to additively manufactured materials. I do have a few questions for the authors.

(1) A large part of the simulations assume equilibrium conditions (Reuss limit). How sure are the authors that their experiments have actually attained equilibrium? I know they show great equivalence between theory and experiment but I was wondering if this is incidental or global.

(2) One of the vexing questions in elastic-plastic deformation studies in a DAC has been the Reuss-Voight interplay and in the past, there has been discussion on how particle size affects this at high packing density and shear. Can the authors comment on this especially in view of their CEA approach.

Reviewer #2 (Remarks to the Author):

The authors present a coupled experimental-analytical and experimental-analytical-computational approach for unraveling stress distributions in compression experiments with diamond anvil cells (DAC). The approach may be applied to a) assessing phase boundaries, b) understanding kinetic boundaries in the s.c. rotational DAC, that is: rotational shear experiments with extreme stress gradients.

The approach is interesting, yet it addresses problems specific to a few types of experiments with the DAC, but it does not apply any longer to DAC experiments in general: The community has turned to compression of single crystal specimens in hydrostatic and nearly hydrostatic media which remove much of the issues addressed by the authors here. Furthermore, Laue diffraction and multicrystal indexation approaches allow for assessing strain of individual crystallites in polycrystals. The use of nearly hydrostatic media like neon and helium is now common practice. Ultrahigh compression experiments on simple metals and ruby show that deviations from hydrostaticity are low up to 40-50 GPa, and for most materials tolerable (within +/- 5 GPa) to beyond 100 GPa. The experiment that the authors show in Fig 1 falls back beyond the diamond cell experiment design that has been established over the past decade.

The approach proposed by the authors has its merits and is useful for some experiments, but it is not sufficiently general nor do the authors present a case of sufficiently general scientific importance that would justify publication in Nat. Com (the alpha-omega boundary in Zr is now quite well assessed). I recommend the authors to consider a more topical journal like J. Synch. Rad. High Pressure Research, Rev. of Sci. Instr.

I add a list of comments jotted down during reading the manuscript, with the hope that they may be helpful for the authors:

. 'However, they could not be [L] [SEP]
. 17 measured. Even measured pressure distribution contains significant error.' (lines 9-12) [L] [SEP]
-> This statement is not quite up to date (nor are the references): Recent Laue diffraction experiments across the phase transition boundary quantify strain. The corresponding stress requires independent assessment of the elastic tensor, of course.

. the most advanced characterization of the pressure conditions in a sample is based [L] [SEP]
. 35 on determining the radial distribution of pressure averaged over the sample thickness using volume [L] [SEP]
. 36 of a crystal cell measured by X-ray diffraction (XRD) and equation of state (EOS) determined [L] [SEP]
. 37 under hydrostatic conditions 4,5,8. [L] [SEP]
-> Again, this is NOT the 'most advanced' characterization! Instead, single crystals of either phase should be used or domains of twinned crystals should be indexed. This is well possible in diamond cells. There are new approaches of multicrystal indexation where the UB matrices of each or most grains are assessed - both for white and monochromatic XRD.

. However, EOS for hydrostatic and nonhydrostatic loadings are [L] [SEP]
. 38 quite different 9-12. [L] [SEP]

-> A nonhydrostatic loading does not actually give different isotherms, only if non-hydrostatic strain and stress remain unquantified, the isotherms appear to deviate
The authors are aware of this, but the wording is confusing.

. More importantly, for the XRD beam along the symmetry axis of the DAC [L] [SEP]
. 39 (axial XRD), crystallographic planes that are almost parallel to the beam contribute to the [L] [SEP]

. 40 measured XRD patterns only, and axial elastic strain $E_{0,zz}$ and consequently stress σ_{zz} do not
. 41 contribute to the pressure, leading to large error (bar over the field variables means averaged over
the sample thickness). In addition, numerous physical, chemical, geological, and mechanical

-> This is true for non-hydrostatic experiments. The alpha-omega transition in Zr is at a pressure that is well within a regime where hydrostatic pressure can be achieved with neon or helium as pressure-transmitting media in a diamond anvil cell. The authors are referring to problems specific to non-optimal experiments or outdated approaches. The authors' approach is useful for analysing data obtained with the s.c. rotational DAC.

. The only paper that claims measurement of radial distribution of all components of the stress tensor is ref. 40.

-> I don't understand this statement. There are numerous published studies on samples compressed in diamond cells under hydrostatic conditions, i.p. on single crystals. This includes studies where elastic tensors were measured with Brillouin spectroscopy along with single crystal diffraction data or axial compressibilities were assessed from single crystal compression data.

The authors address a problem of specific experiments and samples, it is not a general problem and it appears the authors are not aware of much of the recent work. The experimental design shown in Figure 1 shows a sample compressed btw the diamond anvils. This is not standard. Even an experiment where a sample is compressed in a gasket between two anvils is quite outdated or restricted to very particular cases such as the rotational DAC that the authors mention in the manuscript.

. Sample thickness profile and pressure-dependence of the yield strength of phases are
. 84 determined using X-ray absorption and broadening of X-ray peaks.

-> Peak width of Debye fringes (I assume that's what the authors refer to when they talk about 'X-ray peaks') depends on strain and grain size. In the experiment that the authors propose grain size would change with load. I would be interested to know how the authors suggest to discriminate grain size statistics from strain.

The method that the authors propose is potentially useful for the study of polycrystalline aggregates under non-hydrostatic compression. Such conditions occur in a variety of experiments and cannot always be avoided (formation of non-quenchable high pressure phases with laser-heating in diamond cells) or are used intentionally in shear-experiments.

Hence, I believe this paper to be useful for the community of high-pressure experimentalists but the

problem that the authors claim to solve is not a general one. I suggest to submit this paper to a more topical journal than Nature Communications.

Reviewer #1 (Remarks to the Author):

This is a very 'solid' report that attempts to bring several simulations and experimental data together to answer a very long standing question regarding the interplay between stresses, plastic strain and pressure measurement. The authors apply it to the long studied (and probably the problem that raises questions with every study) alpha-omega transition in Zr. The current manuscript also tries to evaluate a plethora of studies that are arising out of shear devices like the rotational diamond anvil cell (RDAC). In fact, this study highlights the pitfalls of studies with RDAC that can end up being highly irreproducible since improper characterization of the stress fields leads to improper estimation of pressure. This is highlighted with simulations and careful experimental data on the radial pressure distributions at various peak pressures that are validated by FEM calculations. I have no hesitation in recommending this for publication as is. This research would be followed by many high pressure researchers from as diverse a field as rock mechanics to additively manufactured materials. I do have a few questions for the authors.

Authors' Response:

We greatly appreciate the Reviewer's positive evaluation of our results.

Reviewer's Comment:

(1) A large part of the simulations assume equilibrium conditions (Reuss limit). How sure are the authors that their experiments have actually attained equilibrium? I know they show great equivalence between theory and experiment but I was wondering if this is incidental or global.

Authors' Response:

We use the Reuss hypothesis for the analytical approach only; FEM does not need it. We added on p. 5:

“This hypothesis appears to work well due to the highly textured polycrystalline aggregate.”

In the limit, when all grains are fully aligned, there is no difference between single and polycrystal, and both Reuss and Voigt hypotheses should work. And yes, the good correspondence between theory and experiment is the main judge of this and other hypotheses for these and many different situations.

Reviewer's Comment:

(2) One of the vexing questions in elastic-plastic deformation studies in a DAC has been the Reuss-Voigt interplay and in the past, there has been discussion on how particle size affects this at high

packing density and shear. Can the authors comment on this especially in view of their CEA approach.

Authors' Response:

We are aware of this discussion. The second Reviewer also raised a question about the effect of the grain size variation on the interpretation of the XRD results. We elaborated our text on p. 4 as follows:

“Importantly, to exclude the effect of strain hardening, change in grain size and dislocation density, and their effect on the thermodynamics and kinetics of PT, we have strongly preliminary deformed Zr until its hardness does not change^{11,35}; grain size and dislocation density in pure α - and ω -Zr do not change with further straining as well⁴⁸.”

Including all these parameters is still possible but requires more advanced modeling and coupling to experiments, which we are working on.

We greatly appreciate the Reviewer's time and efforts in reviewing our paper, useful critical comments, and positive decision.

Reviewer #2 (Remarks to the Author):

Reviewer's Comment:

(1) The authors present a coupled experimental-analytical and experimental-analytical-computational approach for unraveling stress distributions in compression experiments with diamond anvil cells (DAC). The approach may be applied to a) assessing phase boundaries, b) understanding kinetic boundaries in the s.c. rotational DAC, that is: rotational shear experiments with extreme stress gradients.

Authors' Response:

Thank you for the nice summary. We would like to add that this method also allows one to study (a) the plastic flow of various materials under high pressure, (b) the quantitative kinetics of plastic strain-induced phase transformations and chemical reactions, both in DAC and rotational DAC, and (c) initiates high-pressure tribology.

Reviewer's Comment:

(2) The approach is interesting, yet it addresses problems specific to a few types of experiments with the DAC, but it does not apply any longer to DAC experiments in general: The community has turned to compression of single crystal specimens in hydrostatic and nearly hydrostatic media which remove much of the issues addressed by the authors here. Furthermore, Laue diffraction and

multicrystal indexation approaches allow for assessing strain of individual crystallites in polycrystals. The use of nearly hydrostatic media like neon and helium is now common practice.

Authors' Response:

There are no DAC experiments in general. Different designs are used for specific problems, goals, and measurements for hydrostatic and nonhydrostatic experiments. We are aware that there is a community that deeply studies compression of single crystal specimens in hydrostatic and nearly hydrostatic media, utilizing Laue diffraction and multicrystal indexation approaches for assessing the elastic strain of individual crystallites in polycrystals. We communicate with some researchers from this community, trying to find which methods will apply to our problems, and recently have published a joint paper (Nature Communication, 2022, Vol. 13, 982. Here, Laue diffraction alone could not determine the orientation of interfaces between Si I and Si II phases unambiguously; our molecular dynamics and analytical approaches, which showed consistency with the Laue diffraction, resolved this problem and revealed a new nontrivial interfacial nanostructure). It looks like the Reviewer is so deeply, enthusiastically, and successfully involved in this research that he/she missed that numerous other communities lead completely different research driven by practical and fundamental needs. Most engineering materials are polycrystals, and the determination of any property of or processes in a polycrystal based on known properties of and processes in a single crystal (even if we neglect properties of grain boundaries) is a very complex theoretical problem, which is under constant development. Thus, polycrystals should be studied separately, and even when strains of the selected crystallites in polycrystals are measured, this does not give direct answers on the behavior of a polycrystalline aggregate. Also, large communities work on many advanced problems that cannot be studied under hydrostatic conditions. For example, a large geophysical community studies strength and plastic flow under high pressure. A large community uses high-pressure torsion to produce and study nanostructured materials by severe plastic deformations (SPD) and phase transformations. We do the same with traditional and rotational diamond anvils, which allows us to study these processes in situ, combining with FEM simulations. This is unique for nanostructured materials and SPD communities, which is why we are invited to give multiple plenary and keynote lectures at their conferences. The same is true for a large mechanochemical community.

In addition to large communities, there are new emergent directions. For example, contact friction shear stress between diamond and sample/gasket is responsible for generating high pressure, even for quasi-hydrostatic experiments; here, we present the first rules for friction between diamond and Zr. Surface treatment of strong materials (polishing, turning, etc.), deep-focus earthquakes, plastic strain-induced synthesis of (new) materials, the mechanochemical origin of life, etc., involve SPD under high pressure. All of them are mentioned at the beginning and end of the paper, with proper references; all of them cannot be studied under hydrostatic conditions and need methods that we developed in the current paper.

Thus, we do not “address problems specific to a few types of experiments with the DAC;” there are numerous broad problems for which our methods are applicable.

To address the Reviewer’s concern, we added the following text to the paper.

On p. 1:

“We will focus here on stresses and plastic strains averaged over the polycrystalline aggregate rather than in individual grains.”

On p. 2.

“Severe plastic deformation with high-pressure torsion⁴⁵ is an example of the realization of such technologies. Strain-induced PTs under high pressure also occur during surface processing (polishing, turning, scratching, etc.) of strong brittle semiconductors and ceramics and are utilized for developing regimes of ductile machining⁴⁶.”

Reviewer’s Comment:

(3) Ultrahigh compression experiments on simple metals and ruby show that deviations from hydrostaticity are low up to 40-50 GPa, and for most materials tolerable (within +/- 5 GPa) to beyond 100 GPa.

Authors’ Response:

We understand that for the specific goals of those works, such a nonhydrostaticity may be tolerable. But this is not the case for quantitative study of phase transformations. It is known that different pressure-transmitting media, even in the low-pressure range, strongly affect phase transformations, in particular, in Fe [24] and Ti [43], which leads to the large scatter in phase transformation pressures from different works. The problem is not only in nonhydrostatic stresses, but that sample may undergo plastic deformation, leading to a completely different type of phase transformations—strain-induced phase transformations—with completely different thermodynamic and kinetic treatments. There was no way to estimate these effects quantitatively, and our work is the first crucial step toward resolving this general problem. This was written in the concluding part:

“Finding fields of stress tensor components (which can be done using an analytical model) will allow quantifying their effect on the processes under study instead of referring to the qualitative effect of pressure-transmitting media and "non-hydrostatic" stresses. Finding fields of plastic strain tensor components that cannot be measured will allow one to quantitatively study plastic strain-induced PTs and chemical reactions and initiate quantitative high-pressure mechanochemistry.”

Also, ruby shows pressure within a ruby particle, which is different from the pressure in the surrounding sample. The presence of ruby may promote phase transformations by producing a concentration of nonhydrostatic stresses near its surface. That is why we are not using ruby and rely on XRD. A recent Science paper [47], to which we contributed with FEM simulations and general guidance in mechanics, developed a new method to measure normal and shear stresses on the boundary between diamond and pressure-transmitting medium/sample and gasket, again

showing the problem's importance. Our current results are much more ambitious than in [47] because they determine all stress and plastic strain tensor components in the entire sample.

Reviewer's Comment:

(4) The experiment that the authors show in Fig 1 falls back beyond the diamond cell experiment design that has been established over the past decade.

Authors' Response:

The design that the Reviewer referred to is developed for completely different purposes. Design in **Fig. 1** was used in all previous papers [1,4,5,6,21,32,33] in Science, Nature, and PNAS, devoted to determining radial distributions of pressure, thickness, and determination of the pressure dependence of the yield strength. Such a gamut of publications actually shows that our work is not for a specialized journal. For two-stage DACs (like in [2,3,8,9,10]), which are currently used for ultrahigh pressure studies, despite using neon or helium, the sample is compressed directly by anvils (like in **Supplementary Fig. 12c**), and our approach is directly applicable.

Motivated by the Reviewer, we added the following text in p. 11 and **Supplementary Fig. 12**:

“The same methods are applicable for other material systems (including gasket materials), for sample-gasket systems, without and with hydrostatic medium, after its solidification, and can be extended for processes in rotational DAC (**Supplementary Fig. 12**).”

Supplementary Fig. 12: Schematics of DAC assemblies for which the developed approach is applicable. (a) Solid sample within a gasket without a hydrostatic medium (e.g., like in ref.^{18,19} with FEM simulations in²⁰ or any powder material). (b) Solid sample within a gasket with hydrostatic pressure-transmitting medium (PTM) after their solidification (e.g., like in ref.²¹⁻²⁴). (c) The same as in (b) but after the sample is directly compressed by anvils, from the beginning or above some load (e.g., like in ref.^{18,21,25,26}). All fields in the solidified pressure-transmitting

medium and gasket can be studied in the same experiment. The developed approach can be extended for rotational DAC when torque is applied in (a)-(c).

On p. 1.

“The same happens when the pressure-transmitting medium solidifies (**Supplementary Fig. 12**).”

On p. 2.

It is known that phase transformations (PTs) and chemical reactions strongly depend on the nonhydrostatic stresses and plastic strains^{11,16,20,22,23,26-31,41,42}, **even within different pressure-transmitting media at relatively low pressure^{24,43}**.

Reviewer’s Comment:

(5) The approach proposed by the authors has its merits and is useful for some experiments, but it is not sufficiently general nor do the authors present a case of sufficiently general scientific importance that would justify publication in Nat. Com (the alpha-omega boundary in Zr is now quite well assessed). I recommend the authors to consider a more topical journal like J. Synch. Rad. High Pressure Research, Rev. of Sci. Instr.

Authors' Response:

Thanks to the Reviewer's concerns, we believe that we have now convincingly demonstrated that our approach applies to a very broad class of experiments and scientific problems, important for many scientific communities. The fact that particular cases of the same problem have been published in Science, Nature, Nature Communications, and PNAS [1,2,4,5,6,8,9,21,32,33,47] confirms our statement.

Concerning “the alpha-omega boundary in Zr is now quite well assessed,” please see below.

Reviewer’s Comment:

(6) I add a list of comments jotted down during reading the manuscript, with the hope that they may be helpful for the authors:

'However, they could not be measured. Even measured pressure distribution contains significant error.' (Ref. 9-12)

-> This statement is not quite up to date (nor are the references): Recent Laue diffraction experiments across the phase transition boundary quantify strain. The corresponding stress requires independent assessment of the elastic tensor, of course.

the most advanced characterization of the pressure conditions in a sample is based on determining the radial distribution of pressure averaged over the sample thickness using volume of a crystal cell measured by X-ray diffraction (XRD) and equation of state (EOS) determined under hydrostatic conditions^{4,5,8}.

-> Again, this is NOT the 'most advanced' characterization! Instead, single crystals of either phase should be used or domains of twinned crystals should be indexed. This is well possible in diamond cells. There are new approaches of multicrystal indexation where the UB matrices of each or most grains are assessed - both for white and monochromatic XRD.

Authors' Response:

As we already described above, we are not interested in the fields in each grain under quasi-hydrostatic loading, but rather in the heterogeneous fields in the polycrystalline aggregate under nonhydrostatic compression and plastic flow, in line with the cited literature in this field. These are two independent areas, and they should not be confronted with each other.

Reviewer's Comment:

(7) However, EOS for hydrostatic and nonhydrostatic loadings are quite different⁹⁻¹².

-> A nonhydrostatic loading does not actually give different isotherms, only if non-hydrostatic strain and stress remain unquantified, the isotherms appear to deviate

The authors are aware of this, but the wording is confusing.

Authors' Response:

The elasticity rule is the relationship between 6 components of the stress tensor and 6 components of the strain tensor. Formally, any relationship between traces of these tensors, which are pressure and volumetric strain, must depend on the other five components of these tensors, excluding the unphysical case when volumetric and deviatoric responses are fully uncoupled. That is what was quantitatively discussed in Ref. 12-15. Utilization of the hydrostatic EoS for nonhydrostatic loading was the only reason our paper [11] was rejected from PRL. The same problem was raised at many seminars. This was one of the drivers of why we developed the current method.

Reviewer's Comment:

(8) More importantly, for the XRD beam along the symmetry axis of the DAC (axial XRD), crystallographic planes that are almost parallel to the beam contribute to the measured XRD patterns only, and axial elastic strain $E_{0,zz}$ and consequently stress $\bar{\sigma}_{zz}$ do not contribute to the pressure, leading to large error (bar over the field variables means averaged over the sample thickness). In addition, numerous physical, chemical, geological, and mechanical

-> This is true for non-hydrostatic experiments. The alpha-omega transition in Zr is at a pressure that is well within a regime where hydrostatic pressure can be achieved with neon or helium as pressure-transmitting media in a diamond anvil cell. The authors are referring to problems specific to non-optimal experiments or outdated approaches. The authors' approach is useful for analysing data obtained with the s.c. rotational DAC.

Authors' Response:

The maximum hydrostatic pressure at room temperature that can be achieved with He before it solidifies is 12.1 GPa. For higher pressure, the experiment is always nonhydrostatic, which may be tolerated by some communities for some problems and not accepted by other communities working, e.g., on phase transformations. Formally, hydrostatic loading is a very particular case of general loading under the action of all 6 components of the stress tensor and also plastic strain tensors, so our approach is much more general and requires completely different treatments. But of course, it is not needed for pure hydrostatic and close-to-hydrostatic experiments, for which completely different and very important problems are being solved. So, these are completely different fields, which should not be confronted, and we completely and respectfully disagree with the Reviewer's statement about the "non-optimal experiments or outdated approaches."

Yes, the alpha-omega transition in Zr is well studied under hydrostatic pressure, and we use these data (EOS of phases, transformation pressure, and elastic constants) as input data in our approach. But this is only the second paper after [11] on in situ quantitative study of any plastic strain-induced phase transformation, with very different interpretations due to coupling to the theory and FEM simulations. The Reviewer considers nonhydrostaticity as an undesirable effect. As we stressed in the paper and the above responses, numerous problems and processes intentionally involve a combination of high pressure and large plastic deformations, which must be studied. In particular, many dozens of papers study the production of nanograined alpha and omega Zr and their nanocomposite (and many thousands of papers for all other possible materials) by high-pressure torsion. As mentioned above, since we study the same grain refinement and phase transformation process in traditional and rotational DAC in situ, this work attracts significant interest from this large SPD community. Also, we showed that after reaching steady hardness and microstructure, phase transformation pressure is the same in traditional and rotational DACs, which has several consequences. First, for DAC experiments involving plastic deformations, the phase transformations are plastic strain-induced (rather than pressure-induced), which required completely different thermodynamic and kinetic treatment and experimental characterization. Second, one can study nanostructure formation and phase transformation under severe plastic deformations not only in rotational but also in traditional DAC. Of course, rotational DAC allows much broader loading paths, especially at low pressures, which is important for both fundamental and practical applications.

As Reviewer can see in our paper, the alpha-omega phase transformation pressure determined with the developed CEA-FEM approach for plastic strain-induced transformation is 2.7 GPa, 2 times lower than under hydrostatic conditions and even 1.3 times lower than the phase equilibrium pressure, and it is essentially different from 1.36 GPa, which we found based on known EOS method. This pressure is found to be independent of the plastic strain tensor and its path,

particularly of the compression-shear strain path. The kinetics of transformation is completely different than under hydrostatic conditions: time is not a parameter, and plastic strain plays a role of the time-like parameter. The theoretically predicted plastic strain-controlled kinetic equation was verified and quantified; it is found to be independent of the plastic strain at pressures below the initiation of PT and pressure-plastic strain loading path.

These are the first new rules in the relatively new field of plastic strain-induced phase transformations, which has numerous practical applications. Zr is just the first material we study; the same methods apply to any material at much higher pressure, including megabar pressure.

Reviewer's Comment:

(9) The only paper that claims measurement of radial distribution of all components of the stress tensor is ref. 40.

-> I don't understand this statement. There are numerous published studies on samples compressed in diamond cells under hydrostatic conditions, i.p. on single crystals. This includes studies where elastic tensors were measured with Brillouin spectroscopy along with single crystal diffraction data or axial compressibilities were assessed from single crystal compression data.

Authors' Response:

As we already discussed and stressed in the paper, this paper is not about hydrostatic pressure (which does not have distributions because it is homogeneous), single crystals, and elastic moduli. But we use and cite data obtained under hydrostatic pressure and elastic moduli of a Zr single crystal in our simulations.

Reviewer's Comment:

(10) The authors address a problem of specific experiments and samples, it is not a general problem and it appears the authors are not aware of much of the recent work. The experimental design shown in Figure 1 shows a sample compressed btw the diamond anvils. This is not standard. Even an experiment where a sample is compressed in a gasket between two anvils is quite outdated or restricted to very particular cases such as the rotational DAC that the authors mention in the manuscript.

Authors' Response:

We have already addressed this many times, added **Supplementary Fig. 12** with additional designs, and recent references from Nature and Nature Communications journals showing problems for which our method is important.

Reviewer's Comment:

(11) Sample thickness profile and pressure-dependence of the yield strength of phases are determined using X-ray absorption^{4,5,8} and broadening of X-ray peaks^{8,41}.

-> Peak width of Debye fringes (I assume that's what the authors refer to when they talk about 'X-ray peaks') depends on strain and grain size. In the experiment that the authors propose grain size would change with load. I would be interested to know how the authors suggest to discriminate grain size statistics from strain.

Authors' Response:

This is a very important point, and we carefully thought about it. We elaborated our text in p. 4 as follows:

“Importantly, to exclude the effect of strain hardening, change in grain size and dislocation density, and their effect on the thermodynamics and kinetics of PT, we have strongly preliminarily deformed Zr until its hardness does not change^{11,35}; grain size and dislocation density in pure α - and ω -Zr do not change with further straining as well⁴⁸.”

Including all these parameters is still possible but requires more advanced modeling and coupling to experiments, which we are working on.

We also added in “MATERIAL AND EXPERIMENTAL METHODS” Section on p. 12:

“Based on the different angular dependence of the grain size and microstrain contributions to the diffraction peak broadening, they can be separated. The whole powder pattern fitting using the modified Rietveld method (as implemented in MAUD software⁵⁸) was utilized, which takes texturing and stress anisotropy into account.”

Reviewer's Comment:

(12) The method that the authors propose is potentially useful for the study of polycrystalline aggregates under non-hydrostatic compression. Such conditions occur in a variety of experiments and cannot always be avoided (formation of non-quenchable high pressure phases with laser-heating in diamond cells) or are used intentionally in shear-experiments.

Hence, I believe this paper to be useful for the community of high-pressure experimentalists but the problem that the authors claim to solve is not a general one. I suggest to submit this paper to a more topical journal than Nature Communications.

Authors' Response:

We greatly appreciate the Reviewer's time and efforts in examining our paper and challenging critical comments. We took the Reviewer's comments very seriously and added new content to the text and figures. We also described our arguments to the Reviewer about the broadness of the problems that can be treated by our methods, including citing papers from Science, Nature, Nature Communications, and PNAS.

The first Reviewer is very positive about our paper and wrote, “answer a very long standing question regarding the interplay between stresses, plastic strain and pressure measurement” and “This research would be followed by many high pressure researchers from as diverse a field as rock mechanics to additively manufactured materials.”

The current Reviewer did not criticize our specific methods and results. He/she wrote, “The approach is interesting” and “The method that the authors propose is potentially useful for the study of polycrystalline aggregates under non-hydrostatic compression. Such conditions occur in a variety of experiments and cannot always be avoided (formation of non-quenchable high pressure phases with laser-heating in diamond cells) or are used intentionally in shear-experiments. Hence, I believe this paper to be useful for the community of high-pressure experimentalists ...”

The only problem is that the Reviewer is so deeply, enthusiastically, and successfully involved in this research that he/she missed that various completely different large communities lead completely different research driven by practical and fundamental needs, which are addressed in our paper. We would like the Reviewer to imagine that he/she submitted the unique results with Laue diffraction under hydrostatic pressure with strains in selected grains and will get a review that this is not general, and the community has turned to time-resolved dynamic experiments under much higher pressure or to study of the polycrystalline aggregates under general 6 components of the stress and plastic strain tensors. Fighting between different fields will not help in the development of science.

We addressed this misunderstanding in detail and hope the Reviewer will show the broad-mindedness corresponding to his/her high-level status and accept our arguments.

REVIEWERS' COMMENTS

Reviewer #1 (Remarks to the Author):

The authors have answered all my queries and doubts and modified the MS accordingly to reflect my comments. I feel that the MS can be published as is.

Reviewer #2 (Remarks to the Author):

the paper has been revised and addresses all comments of the reviewers. I think this work is technically sound.